# Canagliflozin Promotes Structural and Functional Changes in Proximal Tubular Cell Mitochondria of Hypertensive–Diabetic Mice

**DOI:** 10.3390/ijms262411988

**Published:** 2025-12-12

**Authors:** Mayra Trentin-Sonoda, Yan Burelle, Alex Gutsol, Robert L. Myette, Richard L. Hébert

**Affiliations:** 1Kidney Research Centre, Division of Nephrology, Department of Medicine, Ottawa Hospital Research Institute, Ottawa, ON K1Y 4E9, Canada; mtrentinsonoda@cheo.on.ca; 2Department of Cellular and Molecular Medicine, Faculty of Medicine, University of Ottawa, Ottawa, ON K1H 8M5, Canada; yburell2@uottawa.ca (Y.B.); agutsol@uottawa.ca (A.G.); 3Children’s Hospital of Eastern Ontario Research Institute, Ottawa, ON K1H 8L1, Canada; rmyette@cheo.on.ca

**Keywords:** chronic kidney disease, diabetic kidney disease, SGLT2i

## Abstract

The kidneys have a high-energy demand, relying on great rates of mitochondrial oxidative phosphorylation. Excessive glucose in the tubules leads to defective fatty acid oxidation, playing a key role in tubular injury and diabetic kidney disease progression. Besides its glucose-lowering action, canagliflozin (CANA) promotes kidney protective effects. We aimed to investigate whether the demonstrated kidney protective effects are extended to mitochondrial function and remodeling in proximal tubular cells from hypertensive–diabetic mice. Four weeks after streptozocin (STZ) induction of type 1 diabetes in genetic hypertensive (Lin) mice, they were fed either CANA-infused chow or a regular diet for 1 week. CANA treatment reverted the albuminuric state in LinSTZ mice. In PTECs from male mice, CANA promoted a complex mitochondrial network with less spherical and more branched organelles, with evidence of increased fusion. Those improvements reflected on the mitochondria bioenergetics, where CANA treatment induced an augmented baseline and maximum respiration rate, ATP production, and mitochondria membrane potential in PTECs, compared to LinSTZ. In females, CANA produced a milder response, increasing the mitochondrial network without affecting bioenergetics. In conclusion, in vivo CANA treatment positively affects proximal tubular cells’ mitochondria in male hypertensive–diabetic mice with a minor impact in females. The improvement in mitochondrial function and structure might be key to the kidney-protective effects of CANA.

## 1. Introduction

The pathogenesis of diabetic kidney disease (DKD) includes proximal tubule injury and dysfunction [1]. Hypertension is one of the most common comorbidities associated with diabetes [2]. Together, diabetes and hypertension can worsen the chances of mortality related to cardiovascular complications [3].

The sodium–glucose co-transporter 2 (SGLT2), mainly present in the proximal tubular cells, reabsorbs 90–95% of glucose in normoglycemic conditions [4,5]. SGLT2 inhibitors (SGLT2i, gliflozins), originally developed to target glycemic control [6], also display beneficial effects against cardiovascular diseases (CVDs) [7] and kidney disease progression in patients with or without T2D [8]. A growing number of studies show that SGLT2i can be beneficial in normoglycemic animals with kidney disease [9,10,11]. A similar response is observed in patients where non-diabetic individuals also display a great protection against CVDs and worsening of kidney diseases [12,13]. Our group showed that empagliflozin protects against albuminuria in a type 2 diabetes mouse model [14]. Considering the positive outcomes associated with SGLT2i treatments in animal models and clinical trials, such a class of drugs represents a potential therapy for kidney diseases [8,15].

Thus, a better understanding of the effects of SGLT2i is essential for expanding its clinical applications. For example, it remains unclear whether canagliflozin (CANA) exerts protective effects by solely normalizing blood glucose levels or if other cellular processes could contribute to improving kidney injury and function in hypertensive–diabetic models and non-diabetic models/patients.

The kidneys rely on great rates of mitochondrial oxidative phosphorylation (OXPHOS). At rest, the kidneys are the second-highest oxygen consumers. Proximal tubular cells (PTECs) are highly dependent on fatty acid oxidation (FAO) as an energy source for ATP production [16]; excessive glucose in the tubules leads to overactivation of SGLT2 and defective FAO, which plays a key role in tubular injury and diabetic kidney progression [17].

Normalization of blood glucose by SGLT2i stimulates lipolysis and metabolic shift, followed by an increase in the availability of ketone bodies, a more efficient energy source [18,19]. In type 1 diabetic mice, empagliflozin (EMPA) restored mitochondria mass and decreased expression of fission proteins in the kidney [20]. Similarly, ipragliflozin restored tubular cells, mitochondria morphology, and mitochondrial fission/fusion markers in a high-fat diet model [21]. Clinical studies have focused on mitochondria alterations in endothelial cells and the heart [22,23]. While in vitro studies have also investigated the effects of SGLT2i in the mitochondria activity of kidney cells [24,25,26], it remains unclear whether those findings translate to more complexes biological systems.

Here we hypothesized that CANA treatment protects against early detrimental mitochondrial changes in the proximal tubular cells of a mouse model of concomitant hypertension and diabetes type 1 (LinSTZ).

## 2. Results

### 2.1. CANA Lowers Blood Glucose and Protects Against Albuminuria in Hypertensive–Diabetic Mice

To validate our model of diabetes induced by STZ, we measured blood glucose (BG) levels before and after STZ injections. BG was higher in both LinSTZ and LinSTZ + CANA prior to the CANA diet (Figure 1A,B). At the endpoint, the LinSTZ + CANA group had lower BG levels compared to the previous time point and to the LinSTZ group (Figure 1A,B).

Studies conducted and published by our research group have shown that LinSTZ mice develop severe albuminuria and tubular lesions [27,28]. CANA intervention shows that although serum creatinine levels are not affected (Appendix A), CANA-treated mice have lower albuminuria than hypertensive–diabetic mice receiving a regular diet (Figure 1C,D). mRNA levels of the tubular injury marker KIM-1 only showed a trend of an increase in males LinSTZ (Figure 1E), whereas no trend was observed in females (Figure 1F). No significant changes were seen in the histological injury score (Figure 1G–I).

### 2.2. Mitochondria Morphology and Networking in PTECs

Mitochondria morphology and networking can be closely related to mitochondrial function. The technique applied here consisted of z-stack acquisition followed by deconvolution to improve the resolution and processing by a pipeline resulting in 3D projections of the mitochondrial network (simplified diagram shown in Appendix A).

In males, the number of mitochondria per cell did not differ in either of the LinSTZ groups (Figure 2B). However, mitochondria volume/cell and volume/mitochondria ratios were increased with CANA treatment compared to WT control and LinSTZ (Figure 2C,D). Mitochondria from LinSTZ mice displayed a more spherical morphology than WT and LinSTZ + CANA (Figure 2E). Analyses of network formation showed that CANA treatment induced mitochondria branching and an increase in branch junctions compared to WT and LinSTZ (Figure 2F,G).

In females, the number of mitochondria per cell was similar within all the groups (Figure 2I). Mitochondria volume/cell and volume/mitochondria ratios were higher in LinSTZ + CANA compared to WT (Figure 2J,K). Sphericity was lower in CANA-treated group compared to both WT and LinSTZ (Figure 2L). The number of branches and branch junctions per mitochondria were only elevated in LinSTZ + CANA vs. WT (Figure 2M,N).

### 2.3. CANA Treatment and Mitochondria Fission and Fusion Markers

To further investigate the morphological changes in mitochondria, we evaluated the expression levels of proteins implicated in mitochondria fission and fusion. In males, protein levels of the mitochondria fusion marker (Mifusin-2–MFN2) were downregulated in PTECs of LinSTZ mice compared to WT and LinSTZ + CANA (Figure 3A,B). No differences were found in fission marker Drp1 (Figure 3A,C). PTECs from CANA-treated mice showed higher levels of TOM20 compared to LinSTZ (Figure 3A,D). No differences were observed for females (Figure 3E–H).

### 2.4. CANA Treatment Alters Ratio of TMRE/MT

To investigate whether structural changes were accompanied by variations in membrane potential, we stained PTECs with tetramethylrhodamine, ethyl ester (TMRE, Figure 4A). TMRE is a cell-permeant dye that is only sequestered by mitochondria with intact membrane potential. Data is presented as mitochondria potential (TMRE staining) normalized to MitoTracker staining to account for variations in total mitochondria content across conditions. In males, TMRE/MT was only statistically different in LinSTZ + CANA vs. LinSTZ (Figure 4B). In females, the ratio TMRE/MT was decreased in both LinSTZ and LinSTZ + CANA compared to WT (Figure 4C).

### 2.5. CANA Treatment Improves Mitochondrial Respiration in LinSTZ Mice

Next, we examined whether the observed changes could impact bioenergetics. Oxygen consumption rate (OCR) was monitored in PTECs subjected to the Seahorse MitoStress Test (Figure 5). The MitoStress Test not only monitors baseline metabolism, but also the ability of mitochondria to respond and/or adapt to changes in energetic demand.

In males, baseline respiration rates in LinSTZ cells did not differ significantly from those of WT (Figure 5B). However, maximal respiratory capacity, ATP production, and reserve were reduced in LinSTZ cells (Figure 5C,E) whereas CANA treatment led to a significant increase in baseline, (Figure 5B), ATP-dependent, and maximal respiration (Figure 5C,D). Conversely, this effect was absent in females (Figure 5F–I). PTECs from both treated and untreated LinSTZ displayed impaired maximum respiration rate and reserve capacity (Figure 5H,J).

### 2.6. CANA Treatment Promotes OXPHOS Remodeling in Males

Finally, to better understand the findings in bioenergetics, we analyzed protein levels of the OXPHOS complexes. In males, the abundance of key OXPHOS complex subunits in LinSTZ PTECs was not altered significantly compared to WT (Figure 6A–C,E), except for a rise in CIV (Figure 6D). Treatment of LinSTZ mice with CANA abolished this rise in CIV abundance and triggered a global reduction in the expression OXPHOS complexes (CII, CIII, and a trend in CV) compared to LinSTZ. In females, the response to LinSTZ differed from males and involved a reduced expression of CV in both LinSTZ and LinSTZ + CANA (Figure 6J) with no changes in other OXPHOS complex subunits (Figure 6F–J). A similar trend was observed in kidney cortex lysate (Appendix A).

## 3. Discussion

Our study demonstrated that CANA treatment not only affects parameters that were originally changed in hypertensive–diabetic mice but can also modulate mitochondria morphology and networking. The main findings of our study are as follows: (1) 7 days of CANA treatment was enough to normalize blood glucose and decrease albuminuria in hypertensive–diabetic mice. (2) CANA treatment favors mitochondria fusion and networking in proximal tubular cells of male LinSTZ. (3) CANA treatment improves bioenergetics of PTECs from male LinSTZ. (4) Mitochondria alterations induced by LinSTZ and CANA treatment differ between male and female hypertensive–diabetic mice.

Our group has demonstrated that CANA can improve kidney injury and morphology in hypertensive–diabetic male mice at a more advanced stage of kidney disease [29]. In the present study, we addressed early changes in proximal tubular cells. We show that mice receiving STZ become diabetic, and CANA can revert the hyperglycemic state. A 7-day treatment with CANA was enough to lower albuminuria in LinSTZ mice, supporting other findings in the literature [30,31]. The tubular injury marker KIM-1 and injury scores were not statistically different from the control group; only a trend to increase was present for the LinSTZ group. Those are consistent with the early injury at this stage of the disease, considering the significant injury we observed in the same model at 2 months after STZ injections [28,29].

Mitochondria fragmentation can be a cause or consequence of mitochondria depolarization and dysfunction. Conversely, mitochondria fusion and networking are associated with mechanisms that counteract metabolic insults, helping to preserve cellular integrity [32]. We first started by examining the impact of LinSTZ and CANA on mitochondrial content and morphology. PTEC from male LinSTZ had unchanged cellular mitochondrial content. However, mitochondria were on average smaller and rounder (higher sphericity) suggesting decreased fusion. This is corroborated by immunoblot analysis showing a downregulation of MFN2 in male LinSTZ vs. WT with no difference in Drp1 abundance. CANA treatment increased cellular mitochondrial content and promoted a more complex mitochondrial network with less spherical and more branched organelles. Female LinSTZ PTECs presented no evidence of decreased fusion; individual mitochondrial volume and sphericity were comparable to WT, with no changes in mitochondrial content and mitochondrial network branching. Those results are not remarkable given the divergent pathophysiology between males and females.

Next, we evaluated whether the structural changes impacted mitochondria bioenergetics. In both male and female LinSTZ PTECs, mitochondrial membrane potential was slightly reduced while baseline respiration rates did not differ significantly from WT. These data indicated that mitochondrial bioenergetics were not overtly disrupted under baseline cell culture conditions. Nevertheless, maximal respiratory capacity, ATP production, and spare respiratory capacity were reduced in PTECs from LinSTZ, suggesting the presence of oxidative impairments under maximally stimulated conditions. Treatment of male LinSTZ mice with CANA led to a significant increase in baseline, ATP-dependent, and maximal respiration, pointing to a global stimulation of mitochondrial bioenergetics. Conversely, this effect was absent in females consistent with the sexual dimorphism observed previously [29]. Moreover, Clotet-Freixas et al. showed that primary PTECs isolated from diabetic patients display a sex-dependent metabolic signature [33]. Those intrinsic differences could partially explain the sex-dependent findings.

Hyperfiltration induced by hypertension and increased glucose reabsorption imposes increases in energetic demand on PTECs. Mitochondria are highly dynamic and can respond differently to make up for the required ATP [32]. Modulations of the OXPHOS complexes had been shown to be present in diabetes models [34,35]. Although we did observe an increase in complex IV in PTECs from male LinSTZ, it is not accompanied by the higher respiratory rate seen in the MitoStress Test. Proper assembly of all CIV subunits is essential to the formation of a functional complex and it is not necessarily related to protein abundance of individual subunits. Thus, upregulation of total COX2 content may not necessarily translate into increased levels of assembled and catalytically active CIV.

In females, the only statistically significant difference was in CV expression, downregulated in both CANA-treated and regular-fed mice (Appendix A). A similar trend was observed for most of the complexes. Clotet-Freixas et al. revealed that PTECs from diabetic females accumulate pyruvate, preventing the excessive activity of the TCA cycle [33]. Regulated expression or degradation of the MTCs might be due to protective mechanisms, preventing exacerbation of oxidative stress associated with increased OXPHOS. Alternatively, females could be in a different stage of the disease, considering that studies show that OXPHOS abundance and activity are dynamically regulated throughout the development of diabetic kidney disease [35].

The sex differences presented here and in our previous study with canagliflozin [29] are not unique to our findings. Several other groups demonstrated the impact of intrinsic hormonal or pathophysiological distinctions in males versus females can impact the outcome of treatments or disease progression [33,36,37,38]. Although clinical studies showed analogous effects of SGLT2i on cardiovascular and kidney protection in men and women [39,40,41], women are more susceptible to adverse side effects of SGLT2i, including diabetic ketoacidosis [42]. Our study further emphasizes the importance of including both sexes in studies involving animal models that investigate the mechanisms of SGLT2i in diabetes and CVDs.

We acknowledge the limitations of our model: (a) a small sample size for the kidney injury assessment; (b) females show higher resistance to STZ-induced beta-cell toxicity [36] and to development of cardiovascular diseases. Although we show the same glucose-lowering effects for both sexes, males and females could present different degrees of kidney injury induced by the concomitant hypertension-diabetes insult; (c) the experiments conducted ex vivo do not necessarily reflect the in vivo microenvironment; (d) we do not assume that CANA effects are limited to proximal tubular cells; however, our study was limited to ex vivo PTEC assays; and (e) we cannot extrapolate the conclusions to long-term effects of CANA treatment. Nonetheless, we believe that our data provides further insight into the impact of CANA treatment on the mitochondria of PTECs submitted to injury.

In the future, studies should be conducted to assess the long-term CANA treatment effect on mitochondria and to further investigate the direct effect of CANA in mitochondria remodeling and biogenesis, as well as a broader profiling of the CANA effect on PTEC transcriptome and metabolomics.

In summary, in vivo CANA treatment on hypertensive–diabetic male mice positively affected ex vivo proximal tubular cells mitochondria. Improvements in mitochondria function and structure might be key to the kidney protective effects of CANA.

## 4. Materials and Methods

### 4.1. In Vivo Study

Mice colonies were housed and bred under protocol number CMM-3810 (University of Ottawa Animal Care Committee, approved on 18 May 2022). Mice had free access to food and water, were kept at a controlled temperature, and had a 12 h light/dark cycle. Experimental procedures were approved by the ethics committee (under protocol number CMM-3809, approved on 18 May 2022). The study was conducted according to the guidelines of the Canadian Council on Animal Care.

Hypertensive mice used in this study (TTRhRen or Lin) were originally generated by Prescott and colleagues [43]. The transgene construct contains a modified human pro-renin cDNA sequence and a 3 Kb region of the mouse liver-specific transthyretin promoter.

The Lin colony was housed and bred in the animal facility of the University of Ottawa. A total of 112 mice (male and female) were used in this study. We used 32 wild-type litter mate controls (16 males, 16 females) and 80 Lin (44 males and 36 females). Eight-to-ten weeks-old Lin on FVB/n background were randomly allocated to groups LinSTZ (22 males, 20 females) and LinSTZ + CANA (22 males, 20 females). Figure 7 shows the timeline of interventions. Diabetes was induced by streptozocin (STZ) injections. Lin mice received 5 days of intraperitoneal (i.p.) injections of STZ (50 mg/kg body weight; cat. S0130, Sigma-Aldrich, Oakville, ON, Canada) to induce hyperglycemia via pancreatic beta cell death [27,28,29]. Four weeks post-STZ injections, Lin and LinSTZ mice were fed a regular diet (10% kilocaloric (kCal); Teklad, Mississauga, ON, USA) and LinSTZ + CANA mice were fed a canagliflozin-infused diet (canagliflozin from APExBIO, Houston, TX, USA, cat. A8333 supplemented with 225 ppm, diet prepared by Envigo, Indianapolis, IN, USA) for 7 days. Based on the average food intake, the calculated dose was 30 mg/kg/day.

At the endpoint, euthanasia was induced by either inhalation of CO_2_ followed by decapitation for proximal tubular cell collection or inhalation of 5% isoflurane and exsanguination (cardiac puncture) for blood, urine, and tissue collection. Blood was collected and processed for downstream plasma analyses. Urine samples were snap frozen. Kidneys were collected in 4% PFA for histological analysis, snap frozen in liquid nitrogen for qPCR, or transferred to perfusion solution for proximal tubular cells isolation.

### 4.2. Biochemistry Analysis

Blood samples were collected at the endpoint in heparinized tubes, kept on ice, and centrifuged for 10 min at 5000× *g*. Plasma fractions were transferred to a clean tube and frozen at −80 °C until subsequent analysis. Plasma biochemistry (cholesterol, creatinine, triglycerides, AST, ALT, blood urea nitrogen, and electrolytes) analyses were performed by IDEXX inc. (IDEXX, Montreal, Québec, Canada).

### 4.3. Blood Glucose

Mice were fasted for 4 h prior to blood collection. Conscious mice were subjected to a saphenous bleed, and a drop of blood was used to measure glucose on OneTouch Ultra^®^2 m (LifeScan, Malvern, PA, USA). Glucose was measured prior to STZ injections, 4 weeks after injections, and at endpoint.

### 4.4. Albuminuria

Albumin levels on urine were measured using the Mouse Albumin Elisa Kit (cat. E99-134, Bethyl labs, Montgomery, TX, USA) following the manufacturer’s protocol. Urine creatinine was assayed using EnzyChrom™ Creatinine Assay Kit (BioAssay Systems, Hayward, CA, USA, cat. E2CT-100), Albuminuria was represented as a urinary albumin to creatine ratio.

### 4.5. Injury Score

At the endpoint, a section of the right kidney was stored in 4% paraformaldehyde fixation solution for 24 h, followed by another 24 h in 70% ethanol before being embedded in paraffin, followed by sectioning (3 μm) and staining with periodic-acid Schiff (PAS). Sectioning, paraffin embedding, and staining were performed by the Louise Pelletier Histology Core Facility—University of Ottawa. The sections were visualized under light microscopy using a 20× objective (Axioskop 2 Imager A1, Zeiss, Germany). An injury score was performed in a qualitative manner and classified based on degree of kidney damage using the following scale: 0—no damage, 1—minimal, 2—mild, 3—moderate, 4—marked and 5—severed damage and injury. Two midhilar coronal cross sections of each kidney (i.e., from each animal) were used to collect data from 10 to 15 cortical nonconfluent view fields (i.e., 30–45 measurements per group). Representative areas were analyzed in a blinded manner.

### 4.6. KIM-1 Expression

RNA was isolated from the kidney cortex with an RNeasy Mini kit (Qiagen, Germantown, MD, USA), following the manufacturer’s instructions. Synthesis of complementary cDNA was performed with the kit High-Capacity RNA-to-cDNA (Applied Biosystems, Waltham, MA, USA, cat. 4387406). Real-time qPCR reaction contained 5 ng of cDNA, 0.1 μM F primer, 0.1 μM R primer, and PowerUp™ SYBR™ Green Master Mix for qPCR (Applied Biosystems, cat. A25742). Primer sequences: Forward AAACCAGAGATTCCCACACG, Reverse GTCGTGGGTCTTCCTGTAGC.

### 4.7. Proximal Tubular Cells Isolation

Proximal tubular epithelial cells (PTECs) were isolated from experimental mice (WT, LinSTZ, and LinSTZ + CANA). Isolation was performed as previously described [44]. Briefly, kidneys were placed in cold perfusion solution, containing (in mM) 1.5 CaCl_2_, 5.0 D-glucose, 1.0 MgSO_4_, 24 NaHCO_3_, 105 NaCl, 4.0 Sodium lactate, 2.0 Na_2_HPO_4_, 5.0 KCl, 1.0 L-alanine, 10 N-2-hydroxyethylpiperazine-N-2-ethanesulfonic acid (HEPES), and 0.2% bovine serum albumin (BSA). Kidneys were sectioned in two longitudinal halves, cortices were dissected, minced, and digested in a perfusion solution with the addition of 0.1% collagenase type V (Sigma-Aldrich, St. Louis, MO, USA, cat. C9262) and 0.05% soybean trypsin inhibitor (Sigma-Aldrich, St. Louis, MO, USA, cat. T6522), pH 7.2 at 37 °C. The cortical digestion was passed through a 250 µm sieve, pelleted, and resuspended in 40% Percoll (Sigma-Aldrich, St. Louis, MO, USA, cat. P1644) solution, containing (in mM) 5.0 D-glucose, 10 HEPES, 1.0 MgCl_2_, 120 NaCl, 4.8 KCl, 25 NaHCO_3_, 1.0 NaH_2_PO_4_, 1.0 L-alanine, 1.4 CaCl_2_, 60 U/mL penicillin, and 60 µg/mL streptomycin. The digested product was centrifuged at 18,500× *g* for 30 min. After centrifugation, the 4th layer containing PTECs was aspirated, pelleted, and resuspended in culture media (DMEM/F12—Gibco, Waltham, MA, USA, cat. 31600034 and 21700075). Cells were seeded onto 8-well chambered cover slips (Ibidi, Fitchburg, WI, USA, cat. 80806), 60 mm, or 100 mm dishes coated with Matrigel^®^ Basement Membrane Matrix 1:30 (Corning LifeSciences, Tewksbury, MA, USA, cat. 354234). In the first 24 h, cells were cultured in low glucose DMEM/F12 (1:1) medium containing 10% fetal bovine serum (FBS), defined medium (5 µg/mL insulin transferrin sodium selenate, 50 nM hydrocortisone, 2 nM 3,3′,5-triiodo-L-thyronine (Sigma Aldrich, St. Louis, MO, USA, cat. I1884, H0888, T5516), 100 U/mL penicillin (P), and 100 µg/mL streptomycin (S). After 24 h, cells were grown in low glucose DMEM/F12 + P/S and defined medium. The purity of the PTEC fraction was ensured by size selection throughout the isolation process (sieve and Percoll gradient), as well as by visual morphological verification and by using a defined media (described above).

### 4.8. MitoTracker Staining and Mitochondria Morphology

Mitochondria staining for morphological analysis was performed using live cells stained with MT. PTECs cultured in 8-well chamber slides (Ibidi, Fitchburg, WI, USA) were incubated with 0.5 μM MT for 20 min at 37 °C, washed three times, and kept in DMEM/F12 phenol red-free media (Gibco, Thermo Fisher, Waltham, MA, USA, cat. 21041025). Z-stacks were acquired on a GE DeltaVision Elite (GE Healthcare, Chicago, IL, USA) using a 60× oil immersion objective and Cy5 filter cube at a light intensity of 10%, and exposure time (0.01 s). On average, captured fields contained 10–15 cells. Post-acquisition conservative ratio deconvolution was performed on SoftWoRx v6 (SoftWoRx Rochester, MI, USA). Quantitative analysis of mitochondrial morphology was conducted using the Mitochondria Analyze plugin (version 2.3.1) in the ImageJ software (ImageJ/Fiji software, version 20170530), as previously described [45]. A simplified explanation of the algorithm is provided in Appendix A.

### 4.9. Seahorse MitoStress Test

Cultured PTECs were passed onto 96-well Seahorse XFe96/XF Pro cell culture microplates (10,000 cells/well) coated with Matrigel^®^ Basement Membrane Matrix 1:30 (Corning LifeSciences, Tewksbury, MA, USA, cat. 354234). Eight to ten technical replicates were used per biological replicate. Cells were grown in regular DMEM/F12 + defined media. A Seahorse cartridge was hydrated with calibrant solution (Agilent, Santa Clara, CA, USA, cat. 100840-000) and incubated at 37 °C without CO_2_. After 48 h, cell plates were washed with Seahorse XF medium (Agilent Technologies, Santa Clara, CA, USA, cat. 103575-100). Cartridge wells were loaded to a final concentration of 1.5 μM Oligomycin, 2 μM FCCP, and 0.5 μM rotenone/antimycin. A Seahorse MitoStress Test was performed on default configuration. After completion of assay, cells were fixed in 4% PFA and stained with 1 μg/mL DAPI (Invitrogen, Waltham, MA, USA, cat. D3571). Pictures of DAPI-stained cells were acquired on EVOS Cell Imagining System FL Auto 2 (Thermo Fisher Scientific, Waltham, MA, USA) with a 20× objective, covering 85% of the wells. Stitched images were analyzed using Fiji/Image macro (ImageJ/Fiji software, version 20170530). Oxygen consumption rate was normalized by cell count/well.

### 4.10. Immunoblotting

Protein lysate was obtained from confluent PTECs by addition of RIPA buffer (200 μL/60 mm dish, 400 μL/100 mm dish). Protein concentration was measured with Pierce™ Dilution-Free™ Rapid Gold BCA Protein Assay (Thermo Scientific, Waltham, MA, USA, cat. A A55861). One technical replicate was used per biological replicate. Laemmli sample buffer (6×) was added to 5 μg of PTEC. Next, samples were loaded onto 12% TGX Stain-Free™ FastCast™ gels (Bio-Rad, Mississauga, ON, Canada, cat. 1610184) and submitted to electrophoresis. Gels were activated under UV light (ChemiDoc MP—Bio-Rad). Proteins were transferred to nitrocellulose membranes (Bio-Rad, Mississauga, ON, Canada, cat. 1620112). Following transfer, images of membranes were acquired under UV light on ChemiDoc MP (Bio-Rad, Mississauga, ON, Canada) for stain-free normalization. Membranes were then blocked in Tris-buffered saline (pH 7.6) solution containing 0.1% Tween 20 (TBS-T) and 3% BSA, for 1 h at room temperature. Primary antibodies were used at the following dilutions: Drp1 (Cell Signaling Technologies, Denver, MA, USA, cat. 8570), MFN2 (Cell Signaling, Danvers, MA, USA, cat. 9482), OXPHOS Rodent WB Antibody Cocktail (Abcam, Cambridge, UK, cat. Ab110413), and TOM20 (Cell Signaling, Danvers, MA, USA, cat. 42406). Incubation with primary antibodies was performed overnight at 4 °C. Secondary antibodies donkey anti-mouse (Jackson ImmunoResearch Lab, West Grove, PA, USA, cat. 715-005-151) and Donkey anti-rabbit (Jackson ImmunoResearch Lab, West Grove, PA, USA, cat. 111-005-003) were used at 1:10,000 for 1 h at RT. Chemiluminescence was induced with the addition of SuperSignal Western Pico Plus Chemiluminescent substrate (Thermo Scientific, Waltham, MA, USA, cat. 34580) and detected on ChemiDoc MP (Bio-Rad, Hercules, CA, USA). Band densitometry analyses were performed on Image Lab Software v. 6.1 (Bio-Rad, Hercules, CA, USA). Band density was normalized by stain-free density.

### 4.11. Mitochondrial Membrane Potential

Mitochondrial membrane potential was evaluated in live cells by dual staining with membrane potential-dependent dye tetramethylrhodamine, ethyl ester (TMRE), and the membrane potential-independent dye MitoTracker DeepRed (MT—Cell Signaling, Danvers, MA, USA, cat. 8778). PTECs cultured in 8-well chamber slides (Ibidi, Fitchburg, WI, USA) were incubated with 0.2 μM TMRE and 0.2 μM MT for 20 min at 37 °C. PTECs were washed three times in DMEM/F12 phenol red-free media (Gibco, Waltham, MA, USA, cat. 21041025). Images were taken on EVOS Cell Imagining System FL Auto 2 (Thermo Fisher, Waltham, MA, USA) using 20× objective using TRITC and Cy5 filter cube sets at exposure time (0.02 s). TMRE and MT minimum and mean intensity were measured using Fiji/ImageJ software (ImageJ/Fiji software, version 20170530). A minimum of 50 cells/animal were utilized in the analyses. Data was presented as intensity or TMRE/MT ratio. To facilitate visualization, images were captured as monochromatic, and color masks were applied with ImageJ (ImageJ/Fiji software, version 20170530). green for MitoTracker and red for TMRE.

### 4.12. Statistics

Graphs and statistical analyses were performed using GraphPad Prism (Version 9.3.1, GraphPad Prism, San Diego, CA, USA). Data is presented as mean ± SEM. One-way or ANOVA test was applied for a single time point, followed by Tukey post-test. For multiple time points, mixed-effects analysis Šídák’s multiple comparisons test was applied. *p* < 0.05 was considered statistically significant.

## Figures and Tables

**Figure 1 ijms-26-11988-f001:**
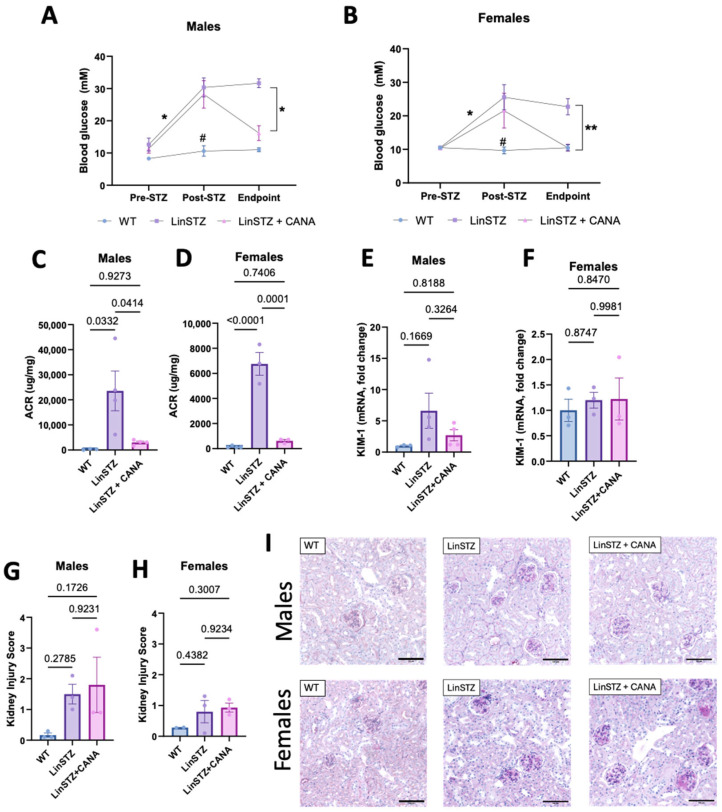
Canagliflozin treatment on blood glucose and kidney injury. Blood glucose levels in (**A**) males and (**B**) females—N = 3 per group per time point. Albuminuria analysis by urinary albumin to creatinine ratio in (**C**) males and (**D**) females. mRNA levels of tubular injury marker KIM-1 in (**E**) males and (**F**) females. Histological analysis of kidney injury by PAS staining in (**G**) males and (**H**) females. (**I**) Representative images of PAS staining (scale bar = 100 μm). Data presented as mean ± SEM. Each dot in the bar graphs represents an animal. N = 2–4 per group. * *p* < 0.05 for pre vs. post-STZ and endpoint LinSTZ vs. LinSTZ + CANA, ** *p* < 0.01 for endpoint LinSTZ vs. CANA, and ^#^
*p* < 0.05 post-STZ measurement WT vs. LinSTZ and LinSTZ + CANA.

**Figure 2 ijms-26-11988-f002:**
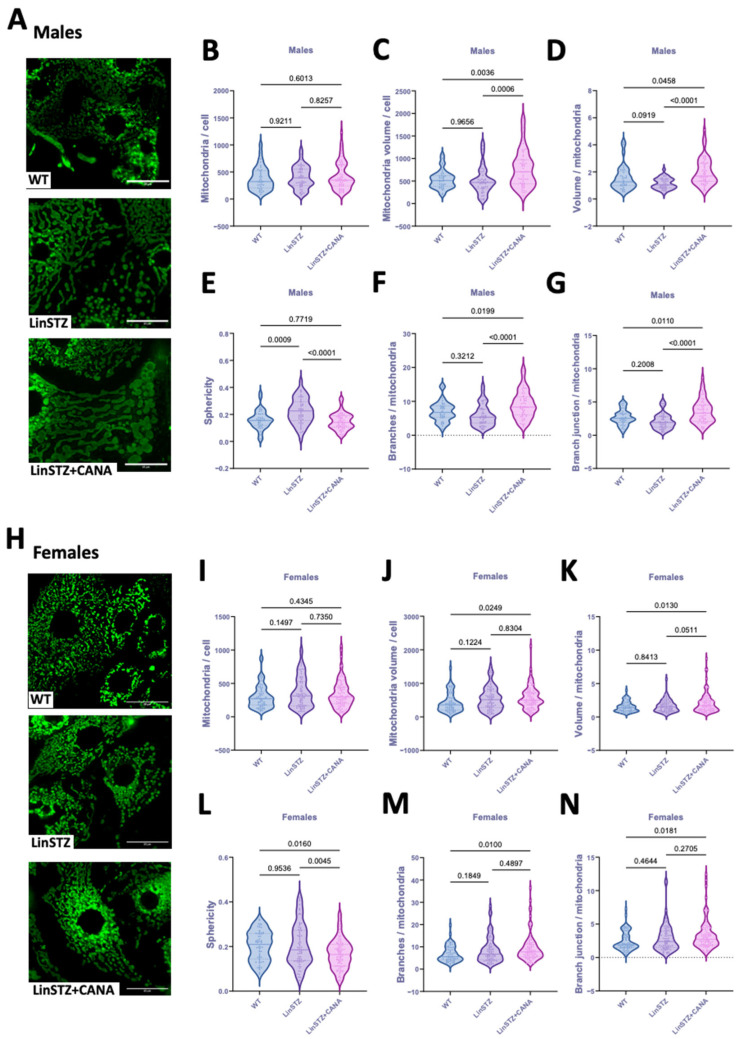
Mitochondria morphology and structure in proximal tubular cells from experimental mice. (**A**) Representative fields of z-stack projection of MitoTracker staining, scale bar = 25 μm. (**B**) Number of mitochondria per cell, (**C**) mitochondria volume per cell, (**D**) volume per mitochondria ratio, (**E**) sphericity, (**F**) number of branches per mitochondria, and (**G**) branch junction per mitochondria in males. (**H**) Representative fields of z-stack projection of MitoTracker staining. (**I**) Number of mitochondria per cell, (**J**) mitochondria volume per cell, (**K**) volume per mitochondria ratio, (**L**) sphericity, (**M**) number of branches per mitochondria, and (**N**) number of branch junctions per mitochondria in females. Data presented as mean ± SEM. Each dot represents a single cell measurement. N = 6–8 animals per group.

**Figure 3 ijms-26-11988-f003:**
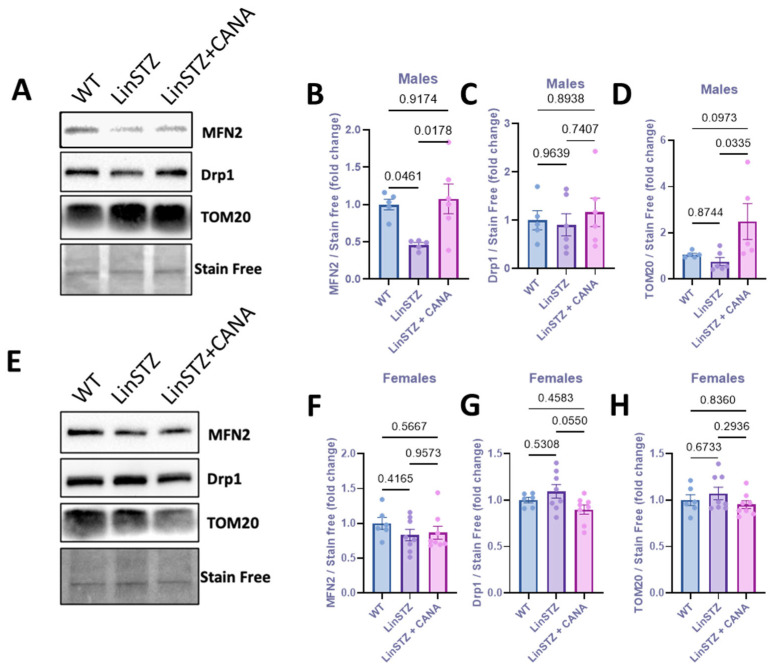
Expression of mitochondria biogenesis proteins in proximal tubular cells from experimental mice. (**A**) Representative immunoblotting image of MFN2, Drp1, and TOM20; fold change relative to the control of (**B**) MFN2, (**C**) Drp1, and (**D**) TOM20 in males. (**E**) Representative immunoblotting image of MFN2, Drp1, and TOM20; fold change relative to the control of (**F**) MFN2, (**G**) Drp1, and (**H**) TOM20 in females. Data presented as mean ± SEM. Each dot represents an experimental animal. N = 6–8 per group.

**Figure 4 ijms-26-11988-f004:**
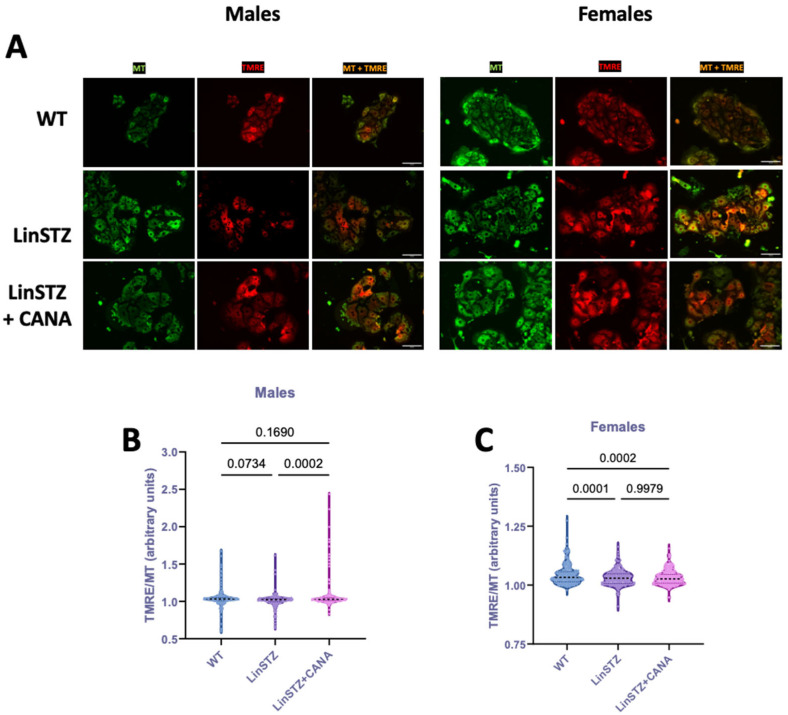
Mitochondria membrane potential in proximal tubular cells (PTECs) of experimental mice. Representative pictures of TMRE (red), MitoTracker (MT, green), and overlap (yellow) live imaging in (**A**) males and females. Scale bar = 25 μm. TMRE/MT ratio in (**B**) male PTECs and (**C**) female PTECs. Data presented as mean ± SEM. Each dot represents a single cell measurement. N = 6–8 animals per group.

**Figure 5 ijms-26-11988-f005:**
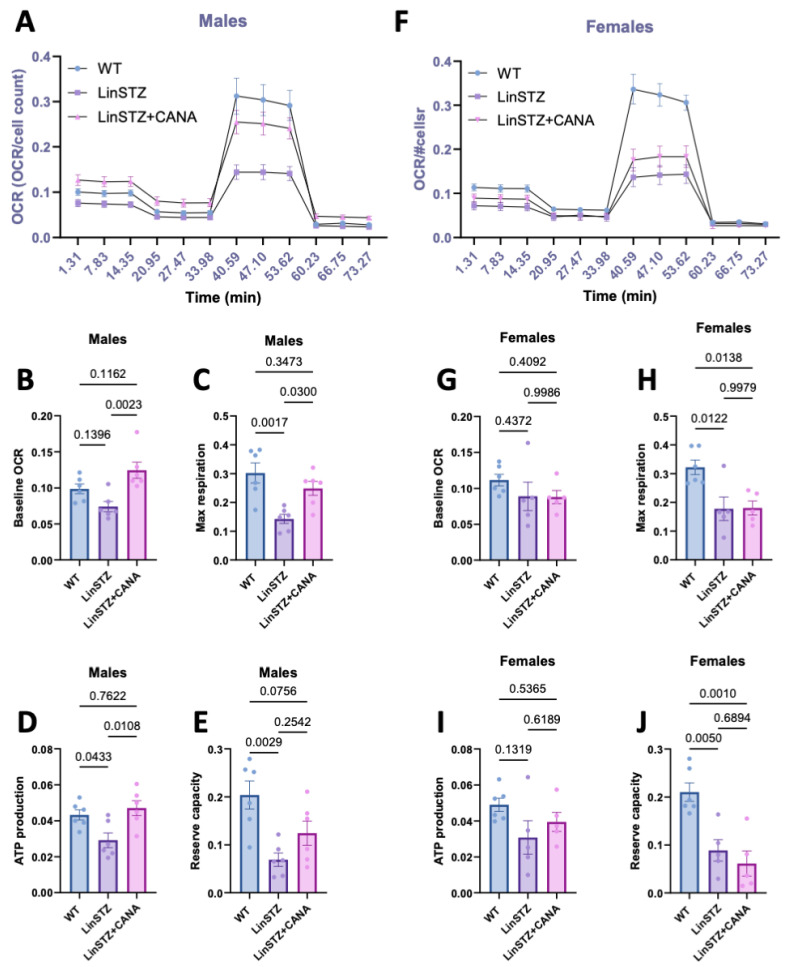
Oxygen consumption rate (OCR) in proximal tubular cells from experimental mice submitted to MitoStress assay. (**A**) OCR overtime, (**B**) baseline respiration rate, (**C**) maximum respiration rate induced by FCCP treatment, (**D**) ATP production, and (**E**) reserve capacity in males. (**F**) OCR overtime, (**G**) baseline respiration, (**H**) maximum respiration induced by FCCP treatment, (**I**) ATP production, and (**J**) reserve capacity in females. Data presented as mean ± SEM. Each dot in the bar graphs represents an animal. N = 6–8 per group.

**Figure 6 ijms-26-11988-f006:**
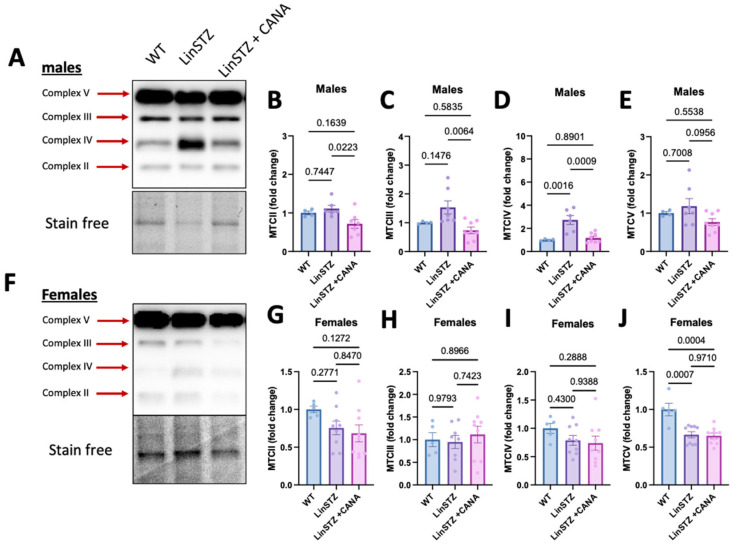
Expression of mitochondria OXPHOS complexes in proximal tubular cells from experimental mice. In males: (**A**) representative image of immunoblotting for (**B**) complex II, (**C**) complex III, (**D**) complex IV, and (**E**) complex V. In females, (**F**) representative image of immunoblotting for (**G**) complex II, (**H**) complex III, (**I**) complex IV, and (**J**) complex V. Data presented as mean ± SEM. Each dot in the bar graphs represents an animal. N = 4–9 per group.

**Figure 7 ijms-26-11988-f007:**
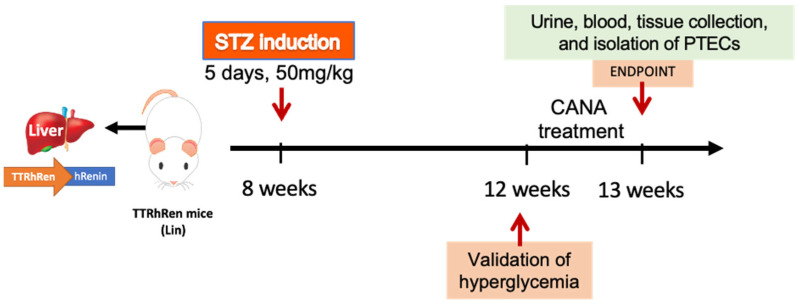
Schematic representation of the timeline of interventions in vivo. Lin mice are treated with STZ for 5 days followed by CANA treatment for 1 week, 4 weeks after STZ injections. At endpoint, urine, blood, and kidneys are collected.

## Data Availability

The original contributions presented in this study are included in the article and Appendix A. Further inquiries can be directed to the corresponding author.

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
