# Peer review of "Canagliflozin Promotes Structural and Functional Changes in Proximal Tubular Cell Mitochondria of Hypertensive–Diabetic Mice"

_ijms, 2025, doi:10.3390/ijms262411988_

Round 1
Reviewer 1 Report
Comments and Suggestions for Authors
This is a well-designed and timely study investigating the effects of canagliflozin (CANA) on mitochondrial structure and function in proximal tubular cells from hypertensive-diabetic mice. The work is methodologically sound, addresses an important gap in understanding SGLT2 inhibitor mechanisms, and highlights sex-dependent responses. The findings are relevant to diabetic kidney disease and may inform future therapeutic strategies. The manuscript is generally clear and well-organized, though some areas could benefit from clarification and expansion.
Major Comments:
- The sex-specific findings are a strength, but the discussion should better contextualize them. The authors mention intrinsic metabolic differences between sexes, but could also discuss hormonal influences, differential expression of SGLT2, or sex-specific mitochondrial adaptations to stress. A more detailed mechanistic speculation would enhance the impact of this observation.
- The 7-day CANA intervention is very short. While it effectively lowered glucose and albuminuria, longer-term effects on mitochondrial remodeling and kidney protection remain unclear. The authors should discuss whether the observed changes are transient or sustained, and whether longer treatment could produce different outcomes, especially in females.
- All mitochondrial analyses were performed ex vivo in isolated PTECs. While informative, these may not fully reflect the in vivo microenvironment (e.g., hemodynamic, hormonal, or inflammatory factors). The authors should acknowledge this as a limitation and consider discussing how in vivo imaging or tissue-based assays could complement future studies.
- The study correlates structural changes with functional improvements in males, but the causal relationship is not explored. For example, does increased fusion directly improve bioenergetics, or is it a consequence of improved metabolic health? Some discussion or speculation on this interplay would be valuable.
Minor Comments:
- The schematics in panel A are helpful but could be more clearly labeled (e.g., “STZ induction,” “CANA diet,” “Endpoint”).
- Sample sizes are small (N=3 per group per time point). Consider mentioning this as a limitation.
The mitochondrial imaging is impressive, but representative images should include insets or higher-magnification views to better illustrate morphological differences.
- In Figure 3, it would be helpful to show loading controls for each blot (e.g., actin or total protein stain).
- Some p-values are missing from bar graphs (e.g., Fig. 1E–I). Ensure all comparisons are clearly indicated.
- Consider adding exact p-values in supplementary tables for transparency.
In the Discussion Section:
- The paragraph on OXPHOS complex changes is somewhat dense. A summary table or schematic in the supplementary material could help readers visualize the sex-specific protein expression patterns.
- Clarify how PTEC purity was confirmed after isolation.
- Specify the number of technical and biological replicates for Seahorse and immunoblotting experiments.
Suggestions for Improvement:
- A graphical summary in the discussion or supplementary material illustrating the proposed mechanism of CANA action on mitochondrial fusion, network complexity, and bioenergetics in males vs. females would be very useful.
- Briefly discuss how these preclinical findings might translate to human DKD, especially in light of known sex differences in CKD progression and SGLT2 inhibitor responses.
- Consider adding a short paragraph on future experiments, such as:
- Longer-term CANA treatment
- Use of mitochondrial fusion/fusion inhibitors to test causality
- In vivo mitochondrial imaging
- Transcriptomic/metabolomic profiling of PTECs
Author Response
Comment : This is a well-designed and timely study investigating the effects of canagliflozin (CANA) on mitochondrial structure and function in proximal tubular cells from hypertensive-diabetic mice. The work is methodologically sound, addresses an important gap in understanding SGLT2 inhibitor mechanisms, and highlights sex-dependent responses. The findings are relevant to diabetic kidney disease and may inform future therapeutic strategies. The manuscript is generally clear and well-organized, though some areas could benefit from clarification and expansion.
Response: We appreciate the reviewer’s positive outlook. The new version of the manuscript was improved by the suggestions made by the reviewers.
Major Comments:
Comment: The sex-specific findings are a strength, but the discussion should better contextualize them. The authors mention intrinsic metabolic differences between sexes, but could also discuss hormonal influences, differential expression of SGLT2, or sex-specific mitochondrial adaptations to stress. A more detailed mechanistic speculation would enhance the impact of this observation.
Response: We acknowledge the importance of such findings. Our decision to carefully word the conclusions on sex differences found with this animal model lays on the fact that the clinical studies do not show any outstanding differences on the effect of SGLT2i between men and women. We have expanded the discussion on sex differences (lines 272-280).
Comment: The 7-day CANA intervention is very short. While it effectively lowered glucose and albuminuria, longer-term effects on mitochondrial remodeling and kidney protection remain unclear. The authors should discuss whether the observed changes are transient or sustained, and whether longer treatment could produce different outcomes, especially in females.
Response: Indeed, we aimed on performing a shorter intervention for this study. Another limitation of the model is the high mortality rate with extending the duration of the study. We have published a separate study where we looked at the impact of CANA a 1-month treatment on kidney morphology and function, we still observe sex differences (mentioned in lines 217-218). We also added a sentence to limitation to address such comment (lines 294-295)
Comment: All mitochondrial analyses were performed ex vivo in isolated PTECs. While informative, these may not fully reflect the in vivo microenvironment (e.g., hemodynamic, hormonal, or inflammatory factors). The authors should acknowledge this as a limitation and consider discussing how in vivo imaging or tissue-based assays could complement future studies.
Response: We agree with the reviewer, removing the cell from the microenvironment might blunt some of the responses we could see in vivo. We have incorporated a sentence in the discussion to reflect this (lines 288-289).
Comment: The study correlates structural changes with functional improvements in males, but the causal relationship is not explored. For example, does increased fusion directly improve bioenergetics, or is it a consequence of improved metabolic health? Some discussion or speculation on this interplay would be valuable.
Response: Improvements in bioenergetics are directed linked to mitochondria structure and network, improved metabolic health would be directed linked to improvement in mitochondria quality control and biogenesis. In lines 227-228 we explain that mitochondria changes in mitochondria structure can be a cause or consequence of mitochondria dysfunction. Whether Canagliflozin plays a role in normalizing ion imbalance that impacts mitochondria health or directly impacts mitochondria remodeling is still unclear. We did not perform any mechanistic studies to address such issues.
Minor Comments:
Comment: The schematics in panel A are helpful but could be more clearly labeled (e.g., “STZ induction,” “CANA diet,” “Endpoint”).
Response: We have changed Figure 7 to address this comment.
Comment: Sample sizes are small (N=3 per group per time point). Consider mentioning this as a limitation.
Response: N=3 was for the model characterization was chosen based on previous analyses in the lab. We acknowledge this could be a limitation and we have added a sentence in the discussion to highlight such limitation (Lines 284-285)
Comment: The mitochondrial imaging is impressive, but representative images should include insets or higher-magnification views to better illustrate morphological differences.
Response: We understand the concern, maybe the pdf generated by the journal did not display the images in high quality. We have resized the representative images for better visualization (Fig. 2).
Comment: In Figure 3, it would be helpful to show loading controls for each blot (e.g., actin or total protein stain).
Response: In Figure 3 we show an Immunoblotting using an antibody cocktail for rodent OXPHOS; All the complexes show up in the same picture, hence the same loading control for all the bands.
Comment: Some p-values are missing from bar graphs (e.g., Fig. 1E–I). Ensure all comparisons are clearly indicated.
Response: We have reviewed the PDF from IJMS submission system and all the P values are present within the graphs.
Comment: Consider adding exact p-values in supplementary tables for transparency.
Response: The exact p value is shown over the respective bars within the graph, except for figure 1A, B and values that <0.0001.
In the Discussion Section:
Comment: The paragraph on OXPHOS complex changes is somewhat dense. A summary table or schematic in the supplementary material could help readers visualize the sex-specific protein expression patterns.
Response: We have added a figure to the supplementary material (Figure S4) to facilitate the understanding of findings related to OXOHOS immunoblotting.
Comment: Clarify how PTEC purity was confirmed after isolation.
Response: This method of isolation has been thorough validated (line 386). Those cells are selected by 1) sieve-mediated size selection of structures during isolation, 2) percoll dradient-size selection, 3) media with defined factors to select proximal tubular cells. A sentence added to the methods section to address this comment (lines 407-409).
Comment: Specify the number of technical and biological replicates for Seahorse and immunoblotting experiments.
Response: We have added the information to the methods section (lines 425 and 441).
Suggestions for Improvement:
Comment: A graphical summary in the discussion or supplementary material illustrating the proposed mechanism of CANA action on mitochondrial fusion, network complexity, and bioenergetics in males vs. females would be very useful.
Response: We have provided the journal with a graphic abstract that summarizes the findings.
Comment: Briefly discuss how these preclinical findings might translate to human DKD, especially in light of known sex differences in CKD progression and SGLT2 inhibitor responses.
Consider adding a short paragraph on future experiments, such as:
- Longer-term CANA treatment
- Use of mitochondrial fusion/fusion inhibitors to test causality
- In vivo mitochondrial imaging
- Transcriptomic/metabolomic profiling of PTECs
Response: We added a future directions paragraph (lines 294-297).
Reviewer 2 Report
Comments and Suggestions for Authors
The article under review investigates the effects of canagliflozin on the structural and functional changes in the mitochondria of proximal tubular cells in hypertensive-diabetic mice. The study is of significant interest due to the growing prevalence of diabetes mellitus and its complications, including kidney disease. The authors have conducted a comprehensive analysis of the effects of canagliflozin on mitochondrial function in the context of hypertension and diabetes. The results of the study may indicate a positive protective effect of the drug on the kidneys. The article is promising and opens up new perspectives for understanding the mechanisms of action of canagliflozin in the treatment of hypertensive-diabetic patients. However, the introduction and conclusion could be expanded to provide a more comprehensive overview of the problem and the significance of the results obtained. The introduction could provide more background information on the role of mitochondria in kidney function, as well as a more detailed overview of the current treatment options for hypertensive-diabetic patients. This would help to better understand the significance of the study's findings. The conclusion could include a more detailed discussion of the potential clinical implications of the study's results. This would help to highlight the practical relevance of the research. The article should provide a clearer and more detailed justification of the positive protective effect of canagliflozin. This could include a more in-depth analysis of the mechanisms by which the drug affects mitochondrial function. It would be beneficial to clarify whether similar effects are observed in healthy mice or if the effects are specific to hypertensive-diabetic mice. This would help to determine whether canagliflozin has any potential side effects or if it is specifically beneficial in the context of hypertension and diabetes.
Author Response
Reviewer 2
Comment: The article under review investigates the effects of canagliflozin on the structural and functional changes in the mitochondria of proximal tubular cells in hypertensive-diabetic mice. The study is of significant interest due to the growing prevalence of diabetes mellitus and its complications, including kidney disease. The authors have conducted a comprehensive analysis of the effects of canagliflozin on mitochondrial function in the context of hypertension and diabetes. The results of the study may indicate a positive protective effect of the drug on the kidneys. The article is promising and opens up new perspectives for understanding the mechanisms of action of canagliflozin in the treatment of hypertensive-diabetic patients. However, the introduction and conclusion could be expanded to provide a more comprehensive overview of the problem and the significance of the results obtained. The introduction could provide more background information on the role of mitochondria in kidney function, as well as a more detailed overview of the current treatment options for hypertensive-diabetic patients. This would help to better understand the significance of the study's findings. The conclusion could include a more detailed discussion of the potential clinical implications of the study's results. This would help to highlight the practical relevance of the research. The article should provide a clearer and more detailed justification of the positive protective effect of canagliflozin. This could include a more in-depth analysis of the mechanisms by which the drug affects mitochondrial function. It would be beneficial to clarify whether similar effects are observed in healthy mice or if the effects are specific to hypertensive-diabetic mice. This would help to determine whether canagliflozin has any potential side effects or if it is specifically beneficial in the context of hypertension and diabetes.
Response: Dear reviewer, we appreciate the comments and believe that the manuscript was improved by the suggestion.
Although the model utilized here focus on hypertensive-diabetic mice, we would like our findings to be the base for future studies modeling diseases where proximal tubular cells are affected on hyper or normoglycemia. We have expanded the introduction to reflect this (l43-46, 54, 67-68. Regarding the clear role of Canagliflozin in mitochondrial health and function, we cannot extrapolate based on the data we have in the manuscript. To further understand the direct role of CANA in mitochondria alterations we would have to test a more controlled environment, where nondiabetic animals and normoglycemic in vitro systems are treated with different mitochondrial inhibitors. Considering the constrain in time and resources, those would have to be explored in the future (sentence added: lines 294-296). Overall, the discussion has been improved following comments of reviewer 1 as well. We have added a paragraph on clinical vs pre-clinical differences lines, as well as expanded the discussion on clinical implication (lines 275-281).